# Exploring the Social Network Structure of Dementia-Friendly Communities in Rural Taiwan: A Qualitative Study

**DOI:** 10.3390/healthcare13182355

**Published:** 2025-09-18

**Authors:** Hsien-Ting Pan, Shu-Ting Chang, Shofang Chang

**Affiliations:** 1Department of Family Medicine, Yuan’s General Hospital, Kaohsiung 802, Taiwan; y2899@yuanhosp.com.tw; 2Department of Hospital & Health Care Administration, Chia-Nan University of Pharmacy & Science, Tainan 717, Taiwan

**Keywords:** dementia-friendly communities (DFC), people with dementia (PWD), qualitative study

## Abstract

**Background/Objectives**: The rising prevalence of dementia presents significant emotional, psychological, and economic challenges for families. Dementia-friendly communities (DFCs) aim to alleviate these burdens by fostering social inclusion and mutual support for people with dementia (PWD) and their caregivers. This study explores the social network structures within DFCs in rural Taiwan, utilizing social network theory as its framework. **Methods**: A qualitative design was employed, involving semi-structured interviews with eleven participants, including eight caregivers and three case managers. Data were analyzed using grounded theory. **Results**: The results indicate that the DFC social network is grounded in reciprocal relationships and mutual support between families, workplaces, and neighborhoods. Durability depends on stable relationships, sustained support systems, and the preservation of local culture. Cultural alignment and engagement reduce stigma and foster understanding, while diverse activities strengthen social bonds and participation. **Conclusions**: The study highlights the importance of government policy, infrastructure, and public awareness in sustaining dementia-friendly environments. The results offer valuable insights for enhancing community design and policy to better support PWD and their families.

## 1. Introduction

The number of people in Taiwan who reach 65 years is growing. A ‘super-aged’ society is expected in 2025 [1]. This demographic shift has significant implications for public health, particularly the growing prevalence of dementia. The World Health Organization (WHO) has emphasized the importance of dementia care as a public health priority, urging nations to implement policies and allocate resources accordingly [2].

Caregivers of people with dementia (PWD) often experience anxiety, anger, depression, and significant stress due to the financial strain, social burdens, and the inability to maintain personal time. Those impact their mental health and ability to provide care [3]. Therefore, both PWD and their caregivers require social network support to alleviate stress. To address these challenges, Taiwan has implemented dementia prevention and care policies, including the establishment of dementia-friendly communities (DFCs). These communities aim to create a supportive environment for PWD and their caregivers by fostering social networks and integrating community organizations, residents, healthcare professionals, and service providers [4]. DFCs promote resource sharing and collaboration, enabling PWD and caregivers to access necessary resources, information, and emotional support. They aim to mitigate the challenges faced by individuals with dementia and their caregivers by fostering an inclusive environment through social network structures.

Although Taiwan has actively promoted DFCs, several challenges and limitations remain. Research has shown the problems, including the unclear role of community connectors, insufficient resource mapping and integration, limited cross-sector collaboration, and the lack of sustainability of services [5,6]. All these hinder the effectiveness of dementia-friendly initiatives. Consequently, many PWD and their caregivers are unable to access appropriate and continuous support, raising concerns about the long-term development and sustainability of dementia-friendly communities in Taiwan [5]. Furthermore, there are significant gaps in community-based support and social interaction. While institutional healthcare environments may provide adequate services, the broader network of community care and meaningful social engagement remains underdeveloped [6]. Hence, it is essential to explore the key factors of DFCs, particularly from the perspectives of PWD and their caregivers.

Social networks provide crucial emotional, instrumental, and informational support, essential for reducing caregiving stress and enhancing the quality of life. Social network theory is a framework for analyzing how relationships among individuals, groups, or organizations shape social structures and influence behaviors [7,8]. It emphasizes the composition of social relationships and the resources they provide offers a framework for understanding community interactions, support systems, and resource utilization [9,10]. Limited research has analyzed DFCs from the perspective of social network theory. Healthcare and community-based services in rural Taiwan are often less developed than in urban centers. Hence, this study investigates the key factors contributing to the social network structure within DFCs from the perspective of caregivers and case managers, using social network theory as a guiding framework. It is expected to provide recommendations to policymakers for enhancing DFCs based on empirical findings.

## 2. Literature Review

### 2.1. Dementia and Dementia Care

Dementia is a neurodegenerative disease affecting memory, thinking, behavior, and daily functioning. Symptoms include memory decline, cognitive impairment, personality changes, and behavioral disturbances. These symptoms can significantly impact interpersonal relationships, work, and daily living [11].

The WHO’s guidelines on reducing cognitive decline and dementia risk highlight the importance of public health strategies [12]. Factors like exercise, cognitive stimulation, and social engagement play a crucial role in prevention and delaying the onset of dementia [13]. In addition, caregivers of PWD face numerous challenges, including psychological stress, financial burdens, and the impact on family life [14]. Effective support systems and resources are crucial for alleviating caregiver stress and improving PWD and the caregivers’ well-being. Effective care for individuals with dementia requires a combination of medical, emotional, and social support, underscoring the need for community-level interventions [15]. Taiwan’s long-term care system has evolved with the implementation of the Long-Term Care Plan 2.0. This policy emphasizes community-based care, with a three-tiered system (A, B, C) providing localized services to meet diverse long-term care needs. Within the framework, specialized dementia care units and designated areas within care facilities have been established to meet the unique needs of PWD [4].

### 2.2. Dementia-Friendly Communities (DFCs)

Countries like Scotland, the UK, Australia, the US, Japan, and Germany have implemented various approaches to developing dementia-friendly communities (DFCs). Similar programs globally, such as Scotland’s National Dementia Strategy (2013–2016) and Dementia Friendly America, emphasize the importance of social inclusion, community participation, and infrastructure development for PWD and the caregivers. These approaches emphasize increased social participation, public awareness, legal protections, enhanced healthcare services, and improved living environments [16,17]. Research highlights the importance of DFCs in improving the quality of life for PWD and promoting social inclusion. Reminiscence therapy and community engagement are recognized as valuable approaches in DFC development [16,17].

Taiwan has made efforts to improve dementia care resources, including the “Dementia-Friendly City Project” and support for PWD and the family caregivers. Daycare centers and home care services are available to alleviate caregiver burden and promote social interaction for PWD. DFCs aim to create environments where individuals with dementia feel understood and supported. Initiatives such as the “Dementia Friendly Angels” program in Taiwan have successfully trained community members to provide support and recognize the needs of those with dementia [4]. They strive to create inclusive environments for PWD, promoting understanding and acceptance while reducing discrimination. They prioritize the rights of PWD, offering appropriate facilities and services. Major dimensions of DFCs include friendly participation, residents, environment, and organizations.

### 2.3. DFCs and Social Networks

Community care relies on both formal and informal support networks. Formal networks include healthcare professionals and social workers, while informal networks involve family, friends, and community organizations. Social engagement and strong social networks are crucial for PWD and their caregivers [18]. They aim to build these supportive networks, enhancing quality of life and reducing social isolation.

Social network theory highlights the significance of relationships within a community, focusing on the connections and interactions that facilitate resource exchange [7,8,19]. The structures of social networks include the following: 1. Reciprocity: Mutual support within relationships. 2. Durability: The longevity of relationships. 3. Homogeneity: Similarity among network members. 4. Density: The closeness of connections within the network. These dimensions serve as a framework for analyzing the structural aspects of social networks in DFCs.

Social networks play a crucial role in the well-being of older adults, providing emotional support, companionship, and access to resources. Factors like mobility, health, and social engagement influence the characteristics of these networks [20,21]. Hence, social network theory is suitable for exploring the key factors involved in building the structure of a DFC.

## 3. Materials and Methods

This study employs a qualitative research design to obtain an in-depth understanding of social networks within DFCs. A case study approach was used, with two rural communities in Tainan, Taiwan, serving as the research setting. Case study is a qualitative research approach that provides an in-depth exploration of a bounded system, such as an individual, group, organization, or community, within its real-life context. It allows researchers to investigate complex phenomena through multiple sources of evidence (e.g., interviews, observations, documents), offering rich descriptions and holistic understanding [22]. Our research scope is rural areas in Tainan, Taiwan, including Qigu and Guanmiao. They are primarily agriculture-based communities with low levels of urbanization. The goal is to explore the experiences of caregivers and the interactions between PWD and their communities.

### 3.1. Research Design

This study utilizes social network theory, focusing on the experiences of caregivers through in-depth interviews and focus groups. The theoretical framework draws upon House, Umberson, and Landis’ concept of social networks, considering both structural and processual aspects [19]. The interview guide was developed based on social network theory to understand network structure, relationships, and information flow, aligned with research objectives to effectively address research questions. The instrument was finalized following a content validity review by three with diverse expertise in long-term care and community health. They included a professor from the Department of Senior Service and Health Management at Chia-Nan University of Pharmacy and Science, Taiwan; the head of the Division of Long-Term Care at Tainan Municipal Hospital, Taiwan, overseeing integrated care services such as discharge planning, home medical and nursing care, palliative care, dementia care centers, and community-based support stations; and the director of the Community Service Division at Tainan Municipal Hospital, Taiwan, responsible for community outreach and service delivery in the district hospital setting.

The contents of the questions in the study include the following: 1. How long have you been caring for a person with dementia? 2. What kinds of support have you received from other families, neighbors, or community members in your daily caregiving? 3. What mechanisms or activities exist in the community to encourage mutual assistance among members? 4. During your dementia caregiving journey, what kinds of people or organizations have provided support to you? 5. How do you perceive the impact of these ongoing social relationships on the quality of care? 6. What are the backgrounds of participants in the community activities you have joined? How do these backgrounds influence your willingness to participate? 7. How would you describe your interactions with other caregivers or supporters in the community? Could you give some examples? 8. What do you think of the channels provided by the community for caregivers to connect and communicate with each other?

Semi-structured interviews were conducted, which is an open and guided method that allows interviewees to express their thoughts and feelings more naturally and genuinely in a relaxed atmosphere. Compared to unstructured interviews that lack an interview outline, semi-structured interviews can better align with our research topic and are more reliable [23].

### 3.2. Sampling and Participants

Purposive sampling was conducted to investigate the needs of PWD in the community and to formulate relevant dementia-friendly policies through in-depth interviews. In line with the principles of qualitative inquiry, purposive sampling emphasized the recruitment of information-rich participants capable of offering in-depth insights into caregiving experiences and community engagement [24,25,26]. Since PWD and their caregivers are considered a vulnerable population and thus not easily accessible, participants were recruited with the assistance of two directors from hospital community service and long-term care departments and two local community directors. Through these introductions, participants were approached, and their informed consent was obtained prior to participation. This approach ensured ethical recruitment and facilitated the identification of individuals with meaningful caregiving experiences while reducing potential recruitment bias by relying on multiple informants rather than a single referral pathway.

Data collection followed the principle of theoretical saturation. Specifically, after the ninth interview, no new categories emerged regarding the types of community support, barriers to social engagement, or caregiving challenges. To ensure the robustness of this conclusion, two additional interviews (one family caregiver and one case manager) were conducted, which confirmed the recurrence of existing themes without generating new insights. The final sample included 11 participants: six family caregivers, two professional caregivers, and three case managers. Interviews were conducted between November and December 2024, lasting from 30 to 120 min (Table 1).

Prior to each interview, the research purpose, process, and ethical considerations were explained, and informed consent was obtained. To protect participants’ rights and privacy, anonymity and confidentiality were strictly maintained. With participants’ consent, both note-taking and audio-recording were used to ensure the accuracy of the data.

### 3.3. Data Analysis

All the data collected in this study are treated in an anonymous manner. The identity of the participants is represented by abbreviated English letters as participant codes. For example, the first caregiver interviewed was abbreviated as “participant 1,” and this logic was followed for archiving purposes [27]. Verbatim transcripts were prepared immediately after the interviews. Grounded theory was used to analyze the transcript [28]. This approach involves systematic data collection and inductive analysis to develop new theories from observations and data. The data were analyzed using the following grounded theory procedures: 1. Open Coding: Identifying concepts and categories. 2. Axial Coding: Establishing relationships between categories and constructing themes. 3. Selective Coding: Synthesizing themes into a cohesive framework. Therefore, the verbatim transcripts of the 11 interviews were meticulously reviewed to extract core concepts through a process of conceptualization. These concepts were then categorized and linked to participant codes. Subsequently, the categorized concepts were named and organized into overarching themes. Throughout this process, the foundational framework of the study was developed by repeatedly verifying the relationships between themes and categories. Finally, categories that were unrelated to the identified themes were excluded.

### 3.4. Research Trustworthiness

This study used Lincoln and Guba’s trustworthiness as the standard for research quality rigor [26]. Credibility was ensured through researcher and data triangulation [29]. Multiple researchers independently analyzed data to reach consensus. This minimized personal biases and strengthened the study’s internal validity. Care diaries and photographs were also used to support the interview data. Transferability was addressed by providing detailed descriptions of the research context and participants. Dependability was ensured through meticulous data collection and analysis procedures. An audit trail was maintained by systematically documenting every step of data collection, analysis procedures, decision-making processes, and revisions to analytical frameworks. Confirmability was maintained by preserving data, respondent validation, and peer debriefing.

## 4. Results

The data were analyzed based on the thematic framework of social network structure, encompassing the dimensions of reciprocity, durability, homogeneity, and density. Ultimately, the analysis identified four categories under both reciprocity and durability, three categories under homogeneity, and two categories under density. The categories and themes were displayed in Table 2.

### 4.1. Reciprocity

Reciprocity in DFCs involves mutual help between families and residents, community support systems, equitable resource allocation, and work participation.

1.Family and Resident Mutual Help: Family members provide support with daily tasks, while community members offer emotional support to family caregivers and social interaction with PWD.

“We live in a rural area, and our family runs a business here, so there are always customers over 30 to 40 years—although they might have dementia, we’re all very familiar with one another (Participant 3).”

2.Community Support Systems: Access to healthcare services, public transportation, social welfare organizations, and social work services enhances support for PWD and their families.

“As for the competent authorities, the agency in charge of senior learning is the Department of Education, … for the dementia care system, the responsible authority is the Department of Social Welfare, while the care centers fall under the Ministry of Health and Welfare (Participant 9).”

3.Resource Allocation: Equitable resource allocation ensures that residents in rural areas have access to necessary services and opportunities.

“In remote areas, many people move away for work and only come back on weekends. Here, around 30 to 40 percent of the elderly live alone. So when we try to organize family-oriented courses, it can be quite difficult. That’s why we always have to plan ahead and schedule them on weekends or even on holidays (Participant 11).”

4.Work Participation: Encouraging work participation for PWD through friendly workplaces and volunteer opportunities promotes social inclusion and well-being.

“Cleaning or sanitation jobs in the public sectors can be offered with mild dementia. Ideally, the work should be located near their own neighborhoods which provide them familiarity. Jobs at places like libraries, museums, or helping with traffic control at schools are also suitable—after all, in rural areas, there aren’t many cars, so they can take their time (Participant 10).”

### 4.2. Durability

Durability in DFCs is maintained through stable interpersonal relationships, continuous community service provision, cultural activities, and government financial subsidies.

Stable Interpersonal Relationships: Harmonious family relationships, supportive neighborly interactions, and strong bonds between caregivers and PWD contribute to a caring community.

“We strongly encourage seniors to engage with the community, rather than staying at home passively watching television (Participant 8).”

2.Continuous Community Service Provision: Sustainable funding and effective organizational management ensure the ongoing provision of essential services.

“I think there should be more dementia-friendly stores labeled by the government, especially since they’re quite rare in rural areas (Participant 5).”

3.Cultural sustainability of the community: Cultural activities, education, and preservation efforts strengthen community identity and social cohesion.

“We can also encourage PWD to participate in community life through cultural activities like religious festivals, engaging with the general public. It helps stimulate their mind and slow down cognitive decline (Participant 7).”

4.Government Financial Subsidies: Government support for social welfare programs, dementia care centers, and community services ensures the availability of resources.

“For example, here in our rural area, the government provides very limited funding (Participant 4).”

### 4.3. Homogeneity

In DFCs, homogeneity refers to cultural alignment within communities that fosters inclusivity, strengthens cooperative efforts through shared values, encourages community engagement, and contributes to stigma reduction.

1.Cultural alignment: Embracing multiculturalism and promoting understanding between different cultural groups fosters a sense of belonging.

“Professors from the university bring groups of students for enriching interactions with the PWD and their family. They’ve initiated markets and community revitalization projects in the old street, through courses and partnerships with various academic institutions (Participant 3).”

2.Community Engagement: Active participation in community activities and events strengthens social connections and reduces isolation.

“A convenient store like 7-ELEVEN provides training on dementia-related knowledge and response skills, enabling them to recognize potential situations involving individuals with dementia and offer appropriate assistance. The curriculum includes the “observe, inquire, reassure, assist” and the three communication principles of “do not startle, do not rush, and do not harm dignity”(Participant 7).”

3.De-stigmatization: Efforts to eliminate stigma associated with dementia promote acceptance and understanding.

“When the PWD perform on stage, the stigma associated with it won’t be so severe. When we take care of them well, they can actually walk into the community without any issues (Participant 2).”

### 4.4. Density

Density in DFCs relates to high community participation in activities and the multiplexity of activity themes.

1.Community Participation: High levels of participation in community events and activities strengthen social connections and support networks.

“There are several community centers that are doing a great job,... there are five in total. We have ten villages in total, but only five of them own their activity centers, and they have activities and classes from Monday to Friday every week for the PWD and their caregivers (Participant 11).”

2.Multiplexity of the Activities: Offering a wide range of activities caters to diverse interests and needs, promoting engagement and social interaction for PWD and the caregivers.

“Here in Guanmiao, the community put together some more innovative activities that they may not have encountered before, such as coffee gardening. We can arrange for them to try these things and engage their senses. It’s often the case that participating in activities at the center is the only way they’ll have these kinds of experiences, as people here have rare opportunities to try new things (Participant 9).”

The social network of a dementia-friendly community is described as Figure 1.

## 5. Discussion

This study explores the contributing factors to the development of DFCs from a social network perspective, an angle that has not been examined in previous research. The findings may serve as a valuable reference for policy planning and provide a conceptual framework for future large-scale quantitative studies.

This study first highlights the critical role of community awareness and understanding in establishing DFCs. This includes accurate knowledge of dementia, as well as support and respect for PWD and their families. In a community free of dementia stigma and people with close relationships and mutual trust, the willingness of PWD and the family caregivers to participate in community activities is generally higher. This research finding aligns with the observation by Lin & Lewis, which indicates PWD living in supportive community experience increased social inclusion, respect, and autonomy [30]. Community activities not only enhance their sense of integration and control within the community but also expand the selectivity and expectations in their lives, allowing them to feel that they can still contribute and enjoy a rich life. Seth, Diler, & Kirk indicate that dementia is a syndrome that can impair daily living abilities due to cognitive dysfunction and may also lead to emotional changes, hallucinations, confusion, and other abnormal behaviors [31]. When a community can establish an open and inclusive atmosphere where PWD and their family feel accepted and respected, the community can become a more friendly and supportive environment.

Secondly, it is imperative to provide training programs and support for PWD and their caregivers. The support provided to PWD and their family caregivers should extend beyond problem-solving skills and professional knowledge to encompass emotional support, enabling them to better navigate the diverse challenges of daily life. The finding corresponds to Bjørge, Kvaal, & Ulstein, which emphasizes the effect of psychosocial support on caregivers’ treatment of PWD [32]. In addition, our study proposes that caregivers share their care experiences and provide community suggestions to help the community better meet the actual needs of individuals with dementia and their families. Their participation can ensure that friendly community projects are more effective and feasible.

Thirdly, with the promotion of sustained economic subsidies from the government, such as establishing dementia shared care centers or dementia support points in hospitals, the quality and efficiency of dementia care can be improved [33]. The government should ensure that sufficient resources are invested in rural areas to establish and maintain the infrastructure and services needed for DFCs. This may include funding community centers, providing training and support for volunteers, and improving public transportation facilities.

Our findings correspond with core insights from social network theory. The observed reciprocity within communities reflects Coleman’s [8] conceptualization of social capital, where mutual obligations strengthen social ties and resource exchange. Durability, grounded in sustained interpersonal relationships and community cultural continuity, corresponds to Wenger’s [20] description of elder networks as reliable support systems. The emphasis on cultural alignment and de-stigmatization demonstrates how homogeneity can enhance bonding social capital, consistent with Putnam’s [9] framework. Lastly, the dense participation in community activities mirrors Cornwell, Laumann, and Schumm’s [21] findings that higher network density fosters stronger connectedness and resilience among older adults. By situating our empirical findings within these theoretical perspectives, this study contributes not only to dementia care practice but also to advancing the application of social network theory in community health research.

In practice, the community residents should be encouraged to actively participate in related education and advocacy activities to improve their level of awareness of dementia. By establishing care experience sharing sessions, residents can share knowledge, experiences, and opinions, promoting mutual understanding. It is hoped that PWD and caregivers can work with residents to create a warm and supportive environment, which will help improve the quality of life for the entire community.

Furthermore, the community should establish an effective support system, including providing emergency assistance, establishing a friendly care network, and coordinating community resources. Through such a support system, the community can better meet the actual needs of PWD and the caregivers, creating a caring and supportive environment. In terms of hardware facilities and community planning, attention should be paid to building a barrier-free environment, providing easily identifiable signs, and friendly designs. The friendliness of community facilities, including shops, public places, and transportation, can improve the quality of life for individuals with dementia. To sum up, the community should advocate a spirit of integration, encourage the community residents to help each other, and achieve the goal of common care.

## 6. Conclusions

This study demonstrates that the sustainability and effectiveness of DFCs in rural Taiwan depend not only on service provision but, more importantly, on the strength of their underlying social networks. Communities that foster dementia awareness, inclusiveness, and mutual trust can reduce stigma and increase the willingness of PWD and their caregivers to participate in community life. Training programs and psychosocial support are also essential, as they help caregivers manage both practical and emotional challenges while contributing their experiential knowledge to community initiatives.

Furthermore, communities should establish comprehensive support systems that include emergency assistance, dementia-friendly care networks, and coordinated resources. Collaborative engagement among healthcare providers, residents, PWD, and caregivers can create a warm and supportive environment, thereby enhancing overall quality of life. Improvements in physical and social infrastructure—such as barrier-free environments, clear signage, and dementia-friendly design—are equally critical for ensuring accessibility and improving daily living for PWD.

While Taiwan has established dementia shared care centers to coordinate services between hospitals and communities, our study suggests the need for a broader community network. In particular, primary care clinics should be included to enhance early detection and continuity of care, while schools can contribute to intergenerational education and dementia awareness. For rural areas, where resources and infrastructure are limited, additional measures are required. These include expanding telecommunication infrastructure, deploying mobile medical and social service teams, empowering community directors or village offices to act as connectors, providing transportation subsidies or shared-ride services for PWD, and embedding dementia-friendly practices into local cultural and religious activities. Such measures would complement existing policies and help ensure that DFCs are equitable, inclusive, and sustainable across both urban and rural settings.

## 7. Research Limitations and Future Research Recommendations

This study has several limitations. First, as a qualitative study, reflexivity must be considered. Although multiple researchers independently analyzed the data and engaged in peer debriefing to minimize personal bias, the researchers’ professional backgrounds and interpretations may still have influenced the findings. Second, while the participants were mainly caregivers, patient privacy and ethical concerns made participant recruitment challenging. While purposive sampling ensured information-rich cases, participants were recruited through hospital community service, long-term care departments, and local community centers. This reliance on specific recruitment channels may have introduced source bias and limited perspectives from caregivers not connected to these services. Third, although data collection continued until no new themes emerged, the claim of theoretical saturation is based on a relatively small and homogeneous sample, and additional interviews in other contexts may have yielded further insights.

In addition, this study used a case study approach that was conducted in two rural communities of Taiwan; therefore, the findings cannot be generalized to all rural or urban regions. Future research should broaden recruitment strategies, cooperate with community organizations and medical institutions, and explore dementia-friendly models in both rural and urban contexts to enhance transferability and capture more diverse perspectives.

## Figures and Tables

**Figure 1 healthcare-13-02355-f001:**
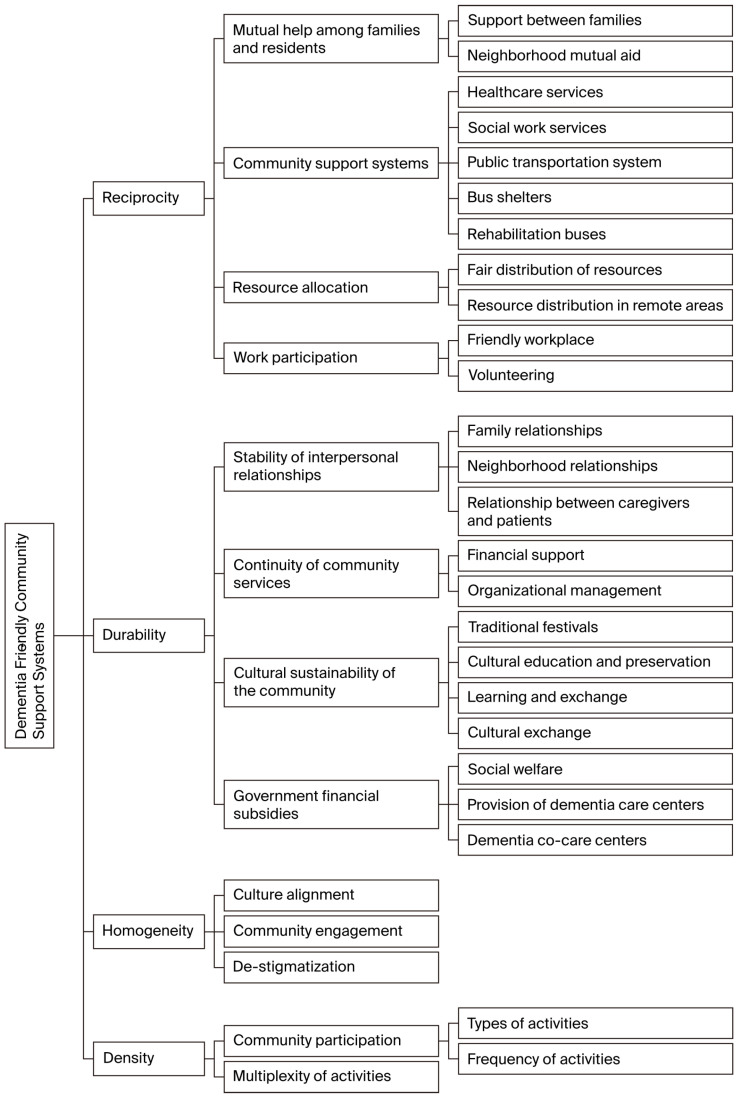
Social network of a dementia friendly community.

**Table 1 healthcare-13-02355-t001:** Demographic characteristics of the participants.

Participant	Role	Caregiving Tenure	Interview Time
Participant 1	Family Caregivers	3 years	48 min
Participant 2		2.5 years	30 min
Participant 3		5 years	73 min
Participant 4		5.5 years	57 min
Participant 5		2 years	54 min
Participant 6		6 years	60 min
Participant 7	Professional Caregivers	8 years	48 min
Participant 8		5 years	45 min
Participant 9	Case managers	5 years	90 min
Participant10		8 years	80 min
Participant11		5.5 years	120 min

**Table 2 healthcare-13-02355-t002:** Categories and themes of the study.

Theme	Category	Concepts
Reciprocity	Mutual help among families and residents	Support between families
		Neighborhood mutual aid
	Community support systems	Healthcare services
		Social work services
		Public transportation system
		Bus shelters
		Rehabilitation buses
	Resource allocation	Fair distribution of resources
		Resource distribution in remote areas
	Work participation	Friendly workplace
		Volunteering
Durability	Stability of interpersonal relationships	Family relationships
		Neighborhood relationships
		Relationship between caregivers and patients
	Continuity of community services	Financial support
		Organizational management
	Cultural sustainability of the community	Traditional festivals
		Cultural education and preservation
		Learning and exchange
		Cultural exchange
	Government financial subsidies	Social welfare
		Provision of dementia care lefts
		Dementia co-care lefts
Homogeneity	Culture alignment	
	Community engagement	
	De-stigmatization	
Density	Community participation	Types of activities
		Frequency of activities
	Multiplexity of activities	

## Data Availability

As some participants in this study belong to potentially vulnerable groups (e.g., caregivers), special attention has been given to protecting their privacy and ethical rights. If access to relevant data is required for the sake of research transparency, requests may be made to the authors. The authors will provide the data only after anonymization and with measures to safeguard participants’ privacy.

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
