# Peer review of "Exploring the Social Network Structure of Dementia-Friendly Communities in Rural Taiwan: A Qualitative Study"

_healthcare, 2025, doi:10.3390/healthcare13182355_

Round 1

Reviewer 1 Report

Comments and Suggestions for Authors

This manuscript explores the social network structures of dementia-friendly communities in rural Taiwan using social network theory. The topic is timely and relevant, and the article offers valuable insights. However, there are several issues the authors need to address before publication:

  1. “People with Dementia” should be consistently abbreviated as “PWD” after first use.
  2. The introduction’s structure is not optimal; organizing it into background, knowledge gap, and study aim would be more suitable.
  3. The authors should clarify how they addressed recruitment bias in participant selection and provide justification for data saturation.
  4. The discussion restates results and compares them to general dementia literature but does not deeply engage with social network theory literature.
  5. The conclusion repeats large portions of the results without providing a clear, high-level takeaway or next-step recommendation.
  6. The limitations section should include reflexivity limitations, recruitment source bias, and theoretical saturation.

Reviewer 2 Report

Comments and Suggestions for Authors

The study exhibits elements that are unclear and that negatively affect its scientific rigour, although it is interesting in itself.

  1. How was the number of participants determined? Specifically, why were there 6 family caregivers, 2 professionals, and 3 case managers? Was it not possible to secure broader participation?
  2. The article refers to ensuring diversity—can this be considered achieved with the given participant data?
  3. The questionnaire was created ad hoc—was it developed solely by the research team, or were other specialists involved in its design?
  4. Was ethical approval obtained from a relevant ethics committee to support the handling of data related to the participants?
  5. Given that this is a qualitative (quasi-experimental) study with a relatively small number of participants, what inferential value can be assigned to the findings? Is there any statistical or other type of element that might support the idea that the results reflect potential social impact? If not, it may be more appropriate to treat this as a case study rather than a broader analysis.
  6. The title suggests a broad study of rural areas in Taiwan, yet the analysis and subsequent limitations indicate that the scope was significantly more limited. It would be advisable to clarify how many areas were analysed and, if applicable, to frame the study as a case study rather than a broader, multi-site analysis.

Round 2

Reviewer 1 Report

Comments and Suggestions for Authors
  1. After first definition of PWD, use “PWD” (no plural “PWDs” or write “people with dementia”.
  2. Typo in Table 2: “Government finanacial subsidies”
  3. IRB approval ID and institution are missing.

Reviewer 2 Report

Comments and Suggestions for Authors

The authors have addressed the methodological issues raised, and the manuscript meets the criteria for publication. 
